# Design and Test of a Crawler-Type Tiger-Nut Combine Harvester

**Zhe Qu, Minghui Han, Yanliu Lv, Zheng Zhou, Zhijun Lv, Wanzhang Wang and Xun He \***

College of Mechanical and Electrical Engineering, Henan Agricultural University, Zhengzhou 450002, China
\* Correspondence: hexun@henau.edu.cn

**Abstract:** Traditional harvesters of tiger nuts (*Cyperus esculentus*) face problems including low harvesting efficiency, high loss rate, high impurity rate, and high labor intensity. To solve these problems and improve the harvesting efficiency and quality of tiger nuts, a crawler-type tiger-nut combine harvester that integrates digging, soil removal, picking, screening, and collection was designed. The machinery comprises crawler devices and working devices. The key devices were designed through theoretical analysis. Therein, the digging and hoisting devices consist of digger blades, combined soil-breaking blades, and vibrating hoisting chains. The tuber picking and screening device is composed of the tuber picking drum, double-deck heterodromous vibrating screens, impurity removal blowers, and soil-crushing guide rollers. The crawler devices include the track assemblies and the hydraulic driving systems. SolidWorks was used to establish the virtual prototype model. Combined with simulation using the discrete element software, the law of motion of tiger-nut tubers in the digging, elevation, and screening processes was studied, which verified the feasibility of the design. Finally, a prototype was manufactured and fabricated to conduct field harvesting tests on tiger nuts. Field test results indicate that the harvesting efficiency, harvest rate, and impurity rate of the tiger-nut harvester are separately 0.216 ha/h, 98.14%, and 3.24%, which meet the harvesting requirements for tiger-nut growers.

**Keywords:** tiger nut; combine harvester; efficient harvesting; mechanical design; field test

## 1. Introduction

Tiger nuts (*Cyperus esculentus*), which are characterized by high yield, absence of occupation of arable land, and tenacious vitality [1–3], have been widely used in edible oil, sugar, grain, and feed industries [4–7]. In recent years, tiger-nut-related industries have developed apace in China [8]; however, the harvesting of tiger nuts is mainly dependent on labor or semi-mechanized means, which shows high labor intensity, high harvesting cost, and extremely low harvesting efficiency, so it cannot meet the demand for the development of tiger-nut-related industries [9,10]. Considering this, it is necessary to develop an efficient combine harvester for tiger nuts to compensate for the deficiency of mechanized harvesting in the tiger-nut-related industries.

In recent years, Liu et al. [11] developed a multi-functional harvester with vibrating screens based on the modular design concept, aiming at the single function of tiger-nut harvesters. Using a tertiary screening process, Zhao et al. [12] studied the impurity removal effect. Zhang et al. [13] investigated the screening devices of tiger nuts and determined the relationship between the screening efficiency and the loss rate. Based on the discrete element method, He et al. [14,15] analyzed the drag reduction of digging devices for tiger nuts, used the digging devices based on reversed rotation and tossing, and improved the digging efficiency. Zhang et al. [16] developed a tiger-nut harvester that can process two different types of tiger nuts in different ways, while the harvesting efficiency is not high. Moreover, [17] studied the effects kinds of threshing rollers—full spike tooth, full-arch tooth, and spike-arch tooth—on the breakage rate and the impurity rate, as well as entrainment loss, by single factor test. The optimal combination of working parameters of the threshing and separating device was obtained. Zhu et al. [18] developed a discrete

element model of flexible plant soil and bionic rotating blades to simulate the process of sedge excavation. Zhao et al. [19] combined the characteristics of the agglomerates of the tiger-net, and a multi-channel transportation device for the tiger-net was designed, to realize the transitional conveyance of the agglomerates of the tiger-net. Existing research on tiger-nut harvesters mainly focuses on traction-type harvesters, while self-propelled tiger-nut harvesters are seldom reported, and there is no self-propelled combine harvester available for tiger nuts in China.

To realize the integrated harvesting of tiger nuts combining digging, soil removal, picking, screening, and collection and then improve the harvesting efficiency and quality, a crawler-type tiger-nut harvester was designed using theoretical analysis, the virtual proto-type technique, and simulation verification based on self-propelled harvesters. Moreover, field tests were conducted to estimate the harvesting efficiency and quality of the prototype, and the overall parameters of the tiger-nut harvester were improved.

## 2. Structure and Working Principle

### 2.1. Planting of Tiger Nuts

Tiger nuts are mainly cultivated through ridge planting or conventional flat planting in some areas [20,21]. The developed crawler-type tiger nut combine harvester is specific for the Yellow River, Huaihe River, and Haihe River Regions in China [22] as shown in Figure 1. The bottom width of ridges, ridge spacing, row spacing on ridges, hill spacing, and ridge height are separately 600 mm, 600 mm, 200 mm, 150 mm, and not lower than 130 mm. Three plants per hill are planted in two rows on each ridge. The depth at which tiger nuts grow is 80 to 120 mm. By investigating the cultivation of tiger nuts, the external dimensions of crawler belts on two sides and the breadth of the digging devices of the proposed tiger nut harvester are both 1800 mm, and the digging depth of the digger blades is adjustable within 0 to 200 mm.

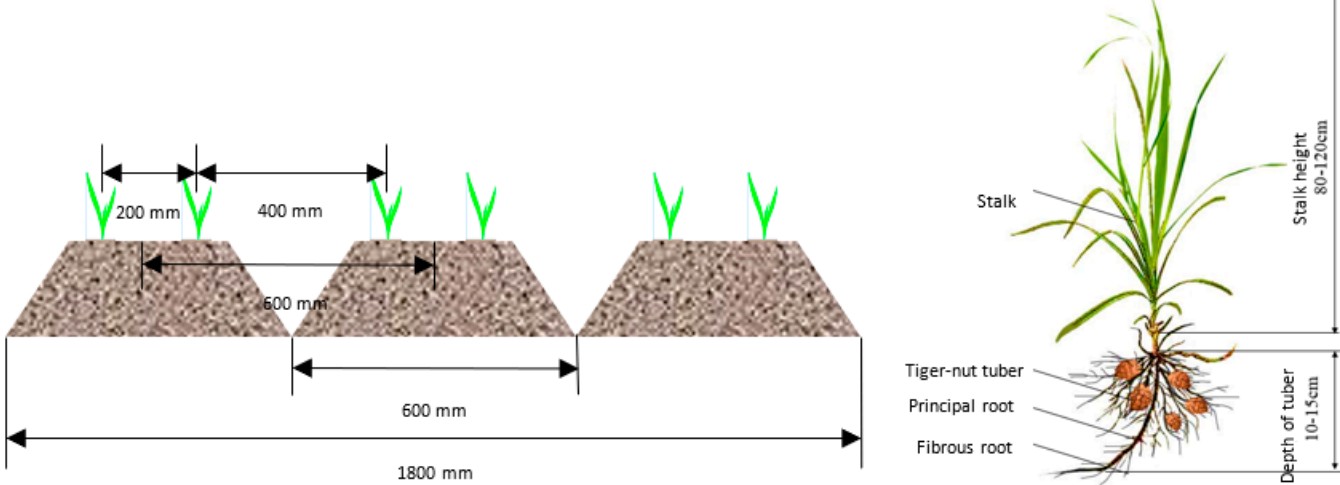

**Figure 1.** Ridge planting of tiger nuts in the Yellow River, Huaihe River, and Haihe River Regions.

The average triaxial dimensions (*L*, *W*, *T*) of 100 undamaged and pest-free tiger nuts were measured with 111N-101 digital display vernier caliper (resolution 0.01 mm). The sizes measured of tiger-nut tubers are illustrated in Figure 2. The results of the triaxial dimensions statistical analysis of tiger nuts are shown in Table 1.

The average length, width, and height of tiger-nut tubers are 11.50 mm, 10.04 mm, and 11.56 mm, respectively. Tiger-nut tubers are sphere-like, with a hundred-grain weight of 83.72 g and density of 1230 kg/m$^3$ [23]. The physical parameters of tiger-nut tubers provide a basis for the design of the screens and tuber picking drum and the establishment of the simulation model.

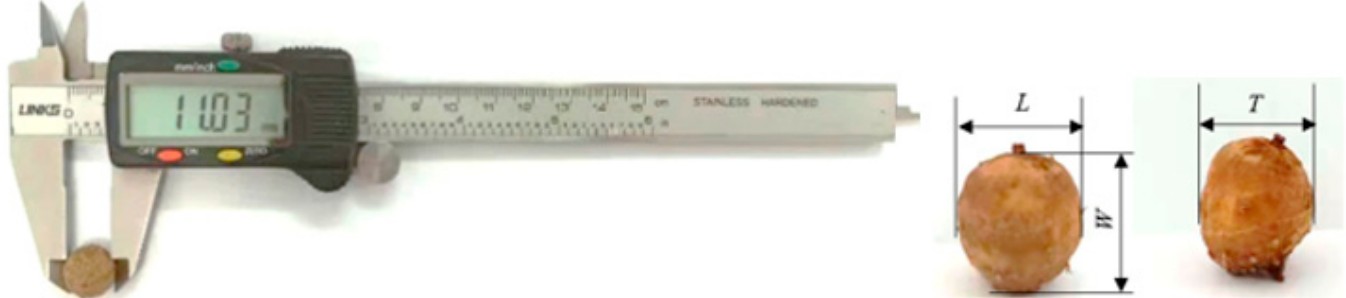

**Figure 2.** Sizes of tiger-nut tubers.

**Table 1.** The results of the triaxial dimensions statistical analysis of tiger nuts.

| Project | Maximum Size | Minimum Value | Mean Value | Standard Deviation |
|---------|-------------|---------------|------------|-------------------|
| $L$(mm) | 14.68 | 8.65 | 11.499 | 1.387 |
| $W$(mm) | 13.59 | 6.61 | 10.044 | 1.351 |
| $T$(mm) | 14.43 | 7.32 | 11.565 | 1.550 |

### 2.2. The Structure

The proposed crawler-type tiger-nut combine harvester is a hydraulically-driven self-propelled machinery, mainly composed of digging and hoisting devices, a tuber picking and screening device, crawler devices, an engine, a driving cab, and a collecting box. It can carry out the combined operation of digging, soil removing, transferring, picking, and collecting tiger-nut tubers in one go. The structure is displayed in Figure 3. The technical parameters of the machine are shown in Table 2.

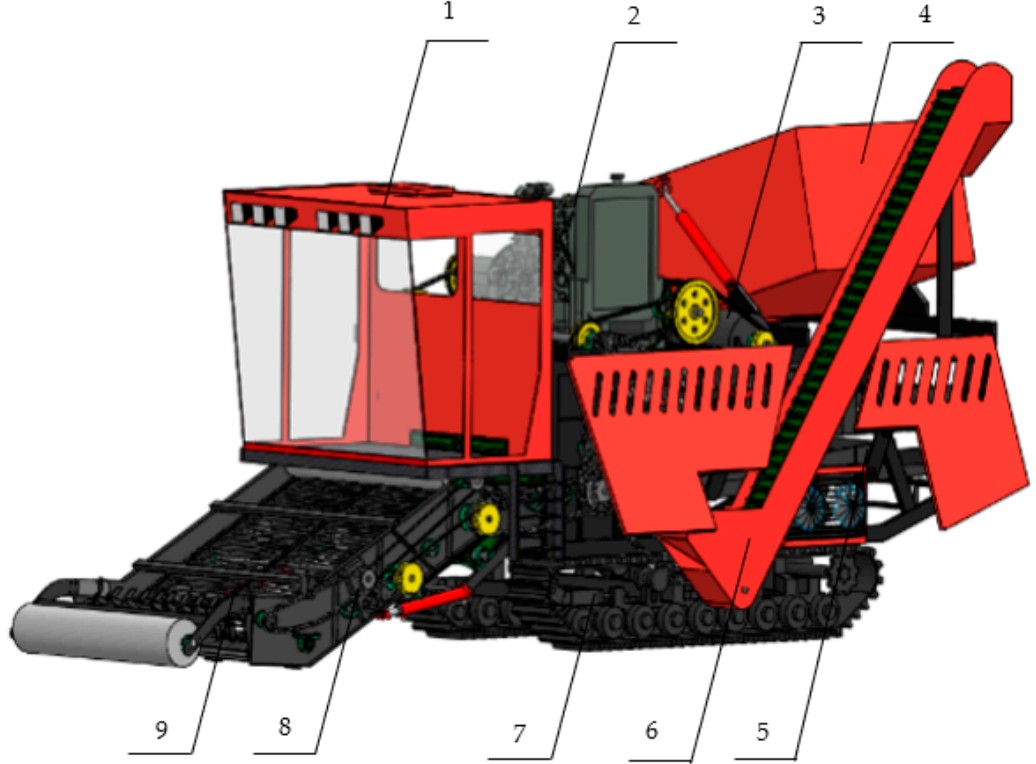

**Figure 3.** The whole structure of the tiger-nut combine harvester. 1. Driving cab; 2. Engine; 3. Tuber picking and screening device; 4. Collecting box; 5. The hydraulic system; 6. Squeegee lift conveyor; 7. Crawler device; 8. Hoisting device; 9. Digging device.

**Table 2.** Parameters table of the tiger-nut combine harvester.

| Item | Value |
| --- | --- |
| Machine size (mm) | $6800 \times 2500 \times 3200$ |
| Engine power (kW) | 140 |
| Digging depth (mm) | 0~200 |
| Number of ridge harvested | 3 |
| Working width (mm) | 1800 |
| Driving speed (m/s) | 0.23~0.33 |
| Harvesting efficiency (ha/h) | 0.15~0.22 |
| Impurity rate (%) | $\leq 5$ |
| Harvest rate (%) | $\geq 95$ |

### 2.3. Working Principle

When the harvester works, the digger blades first excavate the soil burying the tiger nuts. Under the action of soil-breaking blades, the soil containing tiger nuts is crushed and tossed to the hoisting devices. Then, the soil is removed preliminarily from the mixture of tiger-nut tubers, soil particles, and stalks on the hoisting devices and further crushed, followed by transportation of the mixture to the tuber-picking drum. Afterward, the tiger nuts and stalk are further separated in the tuber-picking drum. Finally, after being subjected to reciprocating vibration on the upper vibrating screen, a large amount of soil drops onto the soil dumping plate below and is then discharged from the machine. Light impurities such as stalks are drawn away by the impurity removal blower in the rear of the inlet of the upper screen and discharged out of the harvester. Tiger-nut tubers and remaining soil particles drop from large screen meshes in the rear of the upper screen to the lower vibrating screen to be further sifted. Eventually, clean tiger nuts are obtained and delivered to the collecting box under the action of the tuber delivery device. In this way, the combined harvesting of tiger nuts is realized.

## 3. Design of Key Devices

### 3.1. Design of the Digging and Hoisting Devices

The digging and hoisting devices are mainly adopted to deliver the mixture of tiger nuts and soil to the tuber picking and screening device via processes including digging of digger blades, soil crushing and tossing using soil-breaking blades, transport using hoisting chains, and soil removal, when controlling the digging depth using depth-limiting wheels. To meet the design objectives of improving the soil breaking and tossing performance, increasing the harvesting quality of the whole machinery, and reducing the tuber damage rate, the structure of the digging and hoisting devices was designed, which can be divided into two parts: the digging and the hoisting devices.

According to the planting requirements of tiger nuts, the digging depth was set to be adjustable within 0 to 200 mm and the blade length of triangular digger blades was set to be the same as the harvesting breadth. The penetrating angle of the digging devices is calculated using Equation (1):

$$\alpha = tan^{-1} \frac{F - \mu m}{\mu F + m} \tag{1}$$

where $\alpha$, $F$, and $m$ separately represent the penetrating angle (°), the break-out force for excavation (N), mass of the soil excavated (kg); $\mu$ is the frictional coefficient between digger blades and soil (set to 0.95) [24].

The penetrating angle of the digging devices is calculated to be 15°, and the designed digging devices are displayed in Figure 4. These are composed of triangular digger blades, soil-breaking blades, depth-limiting wheels, and depth-limiting arms. The digging mode based on reversed rotation and tossing is applied and the working breadth is 1800 mm. During operation, the excavation depth is fixed by the depth-limiting wheel 5, the digger blades 2 digs into the soil, and the mixture is sent to the lifting device by the throwing action of the soil-breaking blades 3.

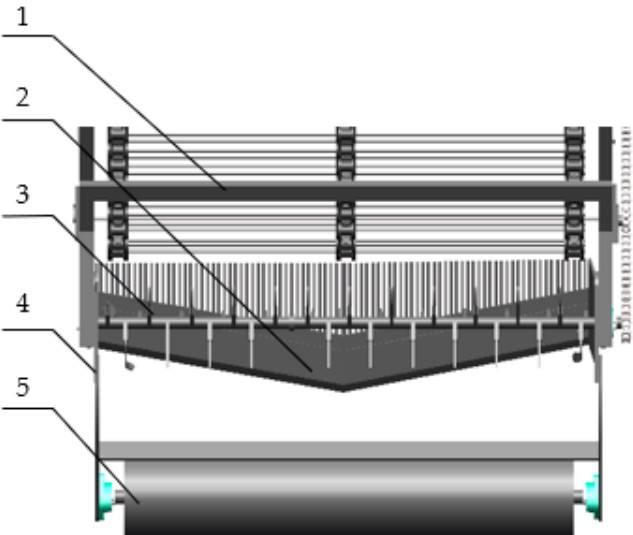

**Figure 4.** The digging devices. 1. Rack; 2. Digger blades; 3. Soil-breaking blades; 4. Depth-limiting arm; 5. Depth-limiting wheel.

The structure of the hoisting devices is illustrated in Figure 5. The devices are mainly composed of a hoisting chain, a vibrating screen, and a vibrating drum. During operation, the mixture of tiger nuts, soil, and grass will be further broken and separated on the hosting surface after being thrown by the digging shovel and the soil-breaking knife. The fine soil will be sieved out of the screen hole and transported to the rear end, so as to reduce the working strength of the rear cleaning and screening device.

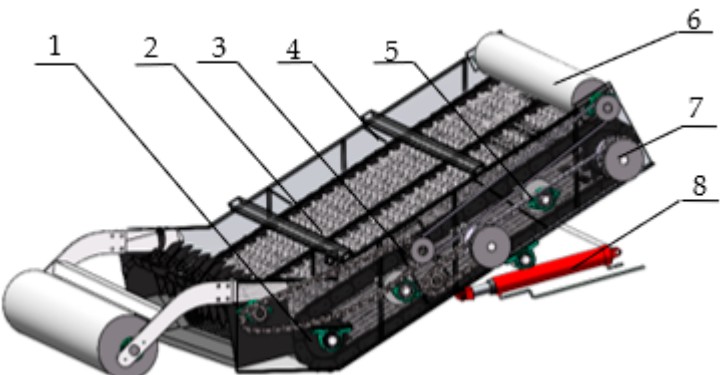

**Figure 5.** The hoisting devices. 1. Driven sprocket; 2. Vibrating screen; 3. Hoisting chain; 4. Vibrating drum; 5. Driving sprocket; 6. Soil roller; 7. Drive sprocket; 8. Hydraulic support cylinder.

According to the physical characteristics of tiger nuts, the aperture of the designed vibrating screens is 7 mm, and the hoisting chain pitch was 63.5 mm. The width of the hosting device is equal to the harvest breadth (1800 mm). A length of 1620 mm can be deduced from the angle calculated by Equation (1).

The stress analysis of the mixture of tiger-nut tubers, soil particles, and stalks in the elevation and motion process on the screen decks is shown in Figure 6 [25]. To enable the hoisting chains to transport the mixture of tiger-nut tubers, soil particles, and stalks upwards instead of dropping, the following condition must be satisfied:

$$P \geq G_1 \sin(\gamma - \varphi) \tag{2}$$

$$f_1 \geq G_1 \cos(\gamma - \varphi) \tag{3}$$

where

$$f_1 = \mu_1 \times N_1 \tag{4}$$

$$P = m' \times \omega^2 \times r \tag{5}$$

$$\omega = \frac{2\pi \times n}{60} \tag{6}$$

$$v = \frac{\omega}{r} \tag{7}$$

Equations (2) to (7) are solved simultaneously to obtain:

$$\omega^2 r = \sqrt{\frac{1}{\mu_1{}^2} + 1} \cdot \sin\left(\gamma - \varphi + tan^{-1}\mu_1\right)g \tag{8}$$

$$\sin(\gamma - \varphi)g \leq \frac{v^2}{r} \leq \sqrt{\frac{1}{\mu_1{}^2} + 1} \cdot \sin\left(\gamma - \varphi + tan^{-1}\mu_1\right)g \tag{9}$$

where $m'$, $\varphi$, and $\gamma$ separately denote the mass of the mixture containing tiger nuts (kg), including the angle between the hoisting devices and the horizontal plane (°), and the intersection angle between the hoisting devices and the horizontal plane (°); $f_1$, $N_1$, $P$, and $G_1$ separately represent the frictional force (N), supporting force (N), centripetal force (N), and gravitational force (N) on the mixture; $n$, $\omega$, $r$, and $v$ denote the rotational speed of the driving wheel of hoisting chains (rpm), the angular velocity of the vibrating drum (rad/min), the radius of the vibrating drum (mm), and the linear velocity of the hoisting chains (m/s), respectively.

According to Equations (8) and (9), influencing factors that relate to the elevation of the mixture of tiger-nut tubers, soil particles, and stalk mainly include the speed and angle of hoisting chains. When the hoisting angle is 15°, it is calculated that the mixture can be elevated smoothly when the linear speed of the hoisting devices is in the range of 0.63 to 0.81 m/s.

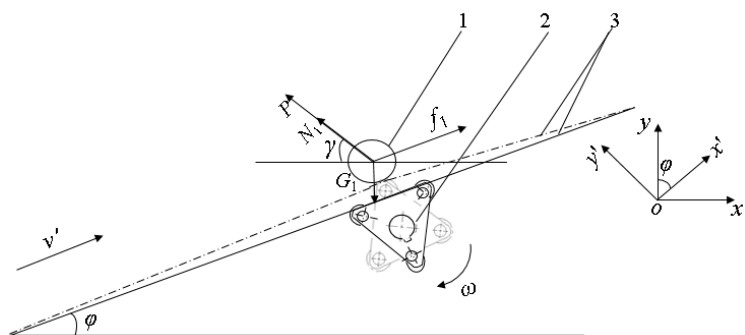

**Figure 6.** Stress analysis of the mixture on the hoisting devices. 1. Mixture of tiger nuts, soil, and stalks; 2. Vibrating drum; 3. Screen. $\varphi$, $\gamma$, and $v'$ separately represent the angle between hosting devices and horizontal plane (°), the intersection angle between the hoisting devices and the horizontal plane (°), and the linear velocity of the hoisting chains (m/s); $f_1$, $N_1$, $P$, and $G_1$ separately represent the frictional force (N), supporting force (N), centripetal force (N), and gravitational force (N) on the mixture, respectively.

### 3.2. Design of the Tuber Picking and Screening Devices

To satisfy the design objective of improving screening efficiency and clean sifting rate, the designed tuber picking and screening device is shown in Figure 7. The device comprises the tuber picking drum, double-deck heterodromous vibrating screen, soil dumping plate, crank-link mechanism, vibrating arm, transverse vibrating screen, impurity removal blower, and transverse conveying screen. The tiger nuts are separated from the stalk in the mixture under the squeezing and flapping of plank teeth and spike teeth in the tuber picking

drum. Then, with the reciprocating vibration of the upper vibrating screen, many small soil particles fall through the front screen mesh onto the soil dumping plate below before being discharged from the machine, while light impurities are drawn out by the impurity removal blower and then discharged. Afterward, the remaining mixture of tiger nuts and soil moves backward under the constant vibration of the screen, and tiger nuts and some soil particles fall from the large screen meshes in the rear section onto the lower vibrating screening. In the meantime, the stems, roots, and stones in the mixture that are heavy and larger than the large screen aperture are directly discharged from the harvester via the rear of the screen. The tiger nuts and some soil falling on the lower vibrating screen move to the front of the machine and the remaining soil is screened. In this way, clean tiger nuts are obtained and delivered to the collecting box via the transverse conveying screen and tuber delivery device.

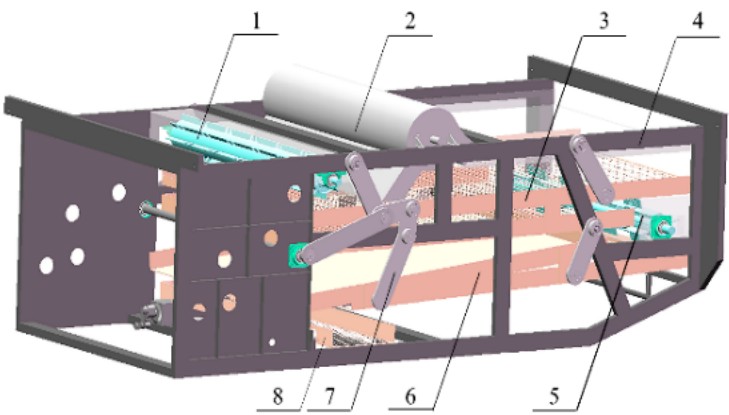

**Figure 7.** The tuber picking and screening device. 1. Tuber picking drum; 2. Impurity removal blower; 3. Upper combined screen; 4. Rack; 5. Soil-crushing guide roller; 6. Lower combined screen; 7. Linkage; 8. Transverse conveying screen.

The tuber-picking drum mainly consists of parts including the closed drum, spike teeth, plank teeth, and shafts. Various components are uniformly distributed on the drum along the spiral line. The threshing and soil crushing are realized under the flapping, rubbing, and squeezing of the spike teeth and plank teeth. The structure of the tuber-picking drum is illustrated in Figure 8.

$$a = z\left(\frac{Z}{K} - 1\right) + 2\Delta_l \tag{10}$$

$$D = d + 2h \tag{11}$$

where $a$ and $z$ separately denote the total length of the drum (mm) and the distance of tooth traces (mm); $Z$ and $K$ represent the numbers of threshing elements and spiral heads respectively; $\Delta_l$ is the distance from the edge tooth to the edge (mm); $D$ and $d$ denote the tip diameter of the drum and the drum diameter (mm), respectively; $h$ refers to the height of threshing elements (mm).

According to the technological requirements imposed for harvesting and screening of tiger nuts, the breadth of the machine, and the physical parameters of tiger nuts, clean tiger-nut tubers with a diameter exceeding 8 mm need to be collected. The structure of the vibrating screens is illustrated in Figure 9. The transverse conveying screen is 1800 mm long. Woven screens are adopted as the vibrating screens. The apertures in the front section of the upper screen, the lower screen, and the transverse screen are all 8 mm, while that in the rear section of the upper screen is 70 mm.

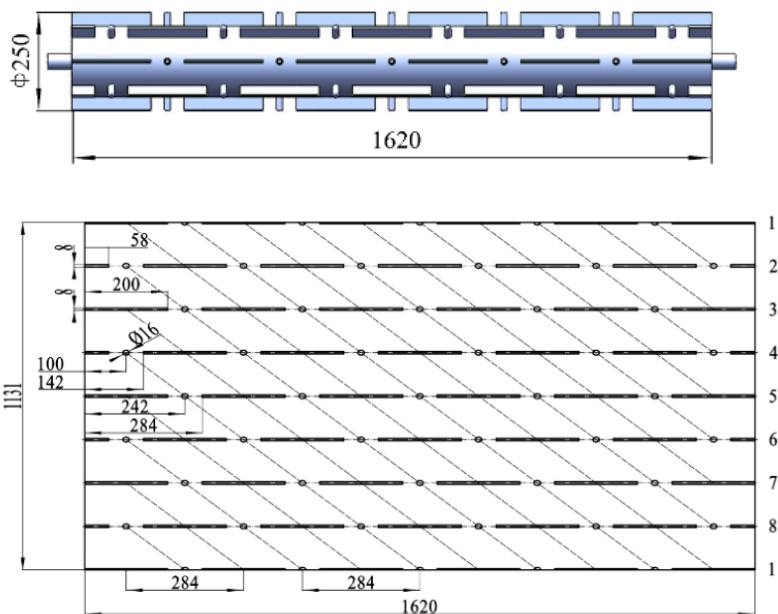

**Figure 8.** The tuber picking drum. Legend: The inclined dotted line represents the spiral line on the drum surface; the intersection between the inclined line and the horizontal line is the location of the central point of threshing elements; the distance of tooth trace between two adjacent spike teeth and plank teeth is 142 mm; the distance between adjacent teeth (the same type of threshing element) is 284 mm.

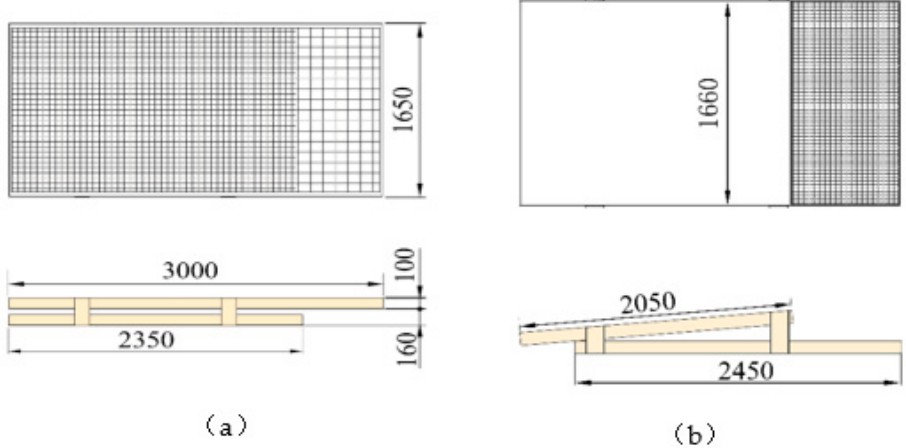

**Figure 9.** Structure of the vibrating screens. (**a**) Model of the upper sieve; (**b**) Model of the lower sieve.

The soil-crushing guide roller installed in the rear of the upper screen can change the motion direction of tiger-nut tubers dropping on the lower screen and preventing tiger nuts from bouncing out of the screen deck from the rear section of the upper screen, which otherwise may cause a loss of yield. In addition, the soil-crushing guide roller can also impact and crush soil for a second time to increase the soil-removal rate. The designed soil-crushing guide roller is shown in Figure 10. It mainly consists of the opened roller, disk, and shafts. The total length of the roller is 1630 mm; the tip diameter is 280 mm, and the length, width, and height of the plank teeth are separately 200, 6, and 60 mm. The plank teeth are uniformly arranged along the spiral line, and the pitch thereof is 718 mm.

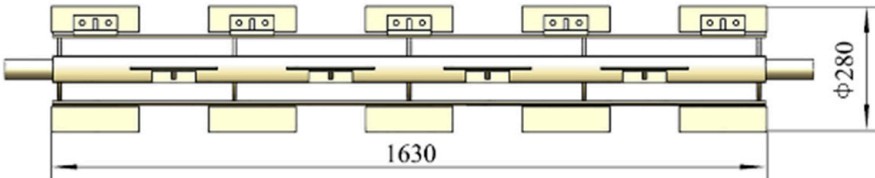

**Figure 10.** Layout of the soil-crushing guide roller and plank teeth.

### 3.3. Design of the Crawler Devices

The crawler devices of the tiger-nut combine harvester are composed of a driving system, left and right track assemblies, and a travelling rack, as displayed in Figure 11. It is used to support the working parts and drive the normal travel of the entire machine.

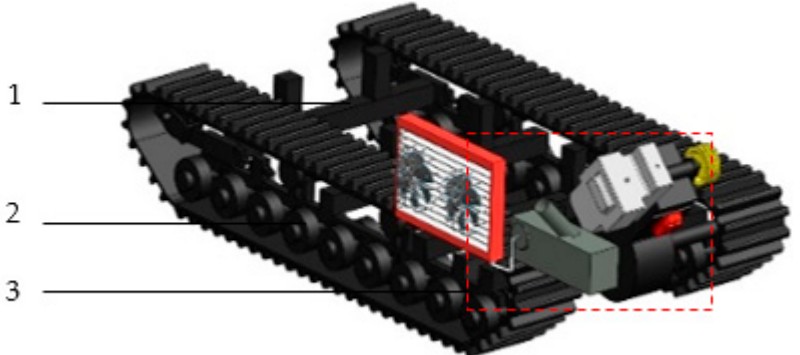

**Figure 11.** Crawler devices. 1. Travelling rack; 2. Left and right track assemblies; 3. Driving system.

In the crawler devices, the stress area of the rubber tracks is large, which can reduce the damage to the soil beneath arising from the excessive weight of the machinery [26,27]. Relevant parameters of the designed rubber track are calculated as follows:

$$L \times b \geq \frac{G}{2[p]} \tag{12}$$

$$\frac{b}{L} = 0.18 \sim 0.2 \tag{13}$$

$$j = (0.027 \sim 0.032)\sqrt[4]{G} \tag{14}$$

$$L' \approx 2L + \frac{z' \times t}{2} + \left(\frac{1}{2} \sim \frac{2}{3}\right)t + 2\Delta \tag{15}$$

$$s = \frac{L'}{j} \tag{16}$$

where $L$, $b$, and $G$ separately denote the length of ground contact of tracks (mm), the width of tracks (mm), and the gravitational force under full loads (kN); $[p]$ represents the permissible specific ground contact pressure; $L'$ signifies the total length of tracks (mm); $z'$ is the number of teeth on the driving wheel (set to nine here); $\Delta$ denotes the pin length (set to 400 mm here); $s$ and $j$ separately represent the number of track pitches and the track pitch (mm).

In the crawler devices, the driving wheel is adopted to wind the track; it transfers power to tracks and provides traction force for traveling. The support wheel can roll freely on the rubber track to support the weight of the harvester and decrease ground impact in motion. The guide wheel guides the winding trajectory of tracks and avoids deviation of the tracks. The tensional device is used to prevent tracks from being slack and derailing. The track roller can constrain the loose side and causes the tracks to sag. The parameters of the aforementioned wheels are calculated as follows:

$$D_o = \frac{j' \times z'}{\pi} \tag{17}$$

$$D_g = D_o - 2F \tag{18}$$

$$D_d = D_o + 2H - 5 \tag{19}$$

$$D_t = 0.8D_o \tag{20}$$

$$D_m = 0.8D_t \tag{21}$$

$$D_j = D_g \tag{22}$$

where $D_o$, $j'$, $D_g$, and $D_d$ separately denote the pitch diameter of the driving wheel (mm), the pitch of the driving wheel (mm), the root diameter of the driving wheel (mm), and the tip diameter of the driving wheel (mm); $H$ represents the thickness of tracks, which is 20 mm here; $D_t$, $D_m$, and $D_j$ denote the diameters of the support wheel (mm), the carrier roller (mm), and the guide wheel (mm), respectively.

The main parameters of the crawler devices are listed in Table 3.

The driving system of the crawler devices is illustrated in Figure 12. When the harvester is running, the hydraulic pump first sucks oil from the oil tank and the swashplate angle of the variable pump is adjusted to change the harvester velocity and control the harvester in its return. Then, the oil drives the hydraulic motor to provide power for the harvester. The oil with dropped pressure after doing work flows back to the oil inlet of the variable pump via the outlet of the hydraulic motor, while the oil leaking in the circuit flows to the low-pressure circuit, followed by topping-up of oil to the variable pump by the slippage pump. After finishing the work, all the oil flows back to the tank. When the harvester is steered, hydraulic oil is pumped by the hydraulic pump from the hydraulic oil tank. Under the control of manual combined directional valves, the machine is steered based on the hydraulic pressure difference of the left and right oil cylinders. When the operating handle is pulled leftwards, the reversing valve moves rightwards and the right side of the hydraulic cylinder protrudes outward, so that the left axle is idle while the right axle is rotated. In this way, the harvester turns left; otherwise, the harvester turns right.

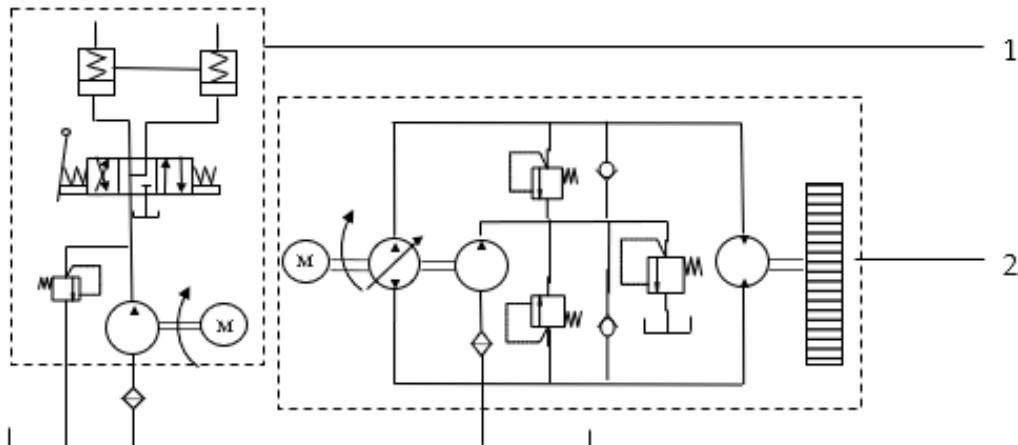

**Figure 12.** Principle of the hydraulic systems. 1. Hydraulic steering system; 2. Hydraulic speed regulating traveling system.

**Table 3.** Main parameters of the crawler devices.

| Item (mm) | Value |
|---|---|
| The length of ground contact of tracks | 2200 |
| The width of tracks | 400 |
| The total length of tracks | 5670 |
| The pitch diameter of the driving wheel | 120 |
| The pitch diameter of the support wheel | 100 |
| The pitch diameter of the guide wheel | 110 |
| The pitch diameter of the carrier roller | 45 |

## 4. Simulation Tests

Based on the above structural design and theoretical analysis, the types of soil-breaking blades were determined and the digging, elevation, picking, and screening processes were numerically simulated to verify whether the designed devices meet practical operating requirements or not.

### 4.1. Establishment of the Simulation Models

Sandy loam is fine and loose and the EDEM (Discrete Element Method) software cannot establish a model according to the actual size of soil particles due to its failure to reach the precision of very tiny particles and limitations in calculation conditions, the soil particles are generally amplified in the simulation. On the premise of not impairing the accuracy of the simulation model, the radius of soil particles was set to 3 mm. In the actual generation of soil bins, soil particles were randomly generated following the normal distribution, to meet the requirement. Tiger-nut tubers which are sphere-like were simulated by combining their dimensions and using the method of building aggregation of spherical particles based on the default contact model (no-slip) in EDEM [28]. Simulation models of the soil particles, tiger-nut tubers, and stalk are shown in Figure 13.

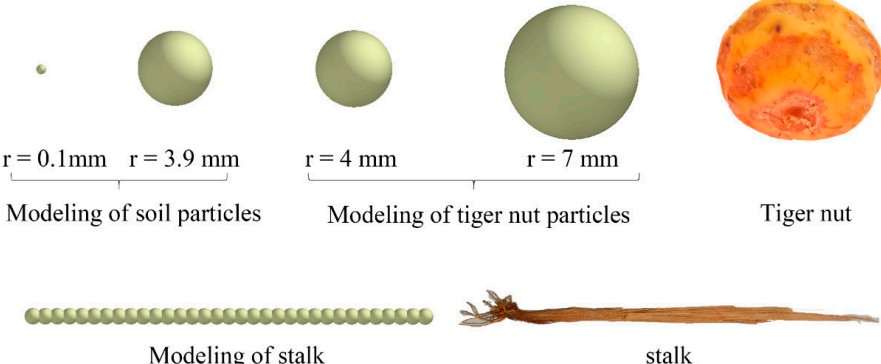

r = 0.1mm   r = 3.9 mm          r = 4 mm              r = 7 mm

Modeling of soil particles      Modeling of tiger nut particles         Tiger nut

Modeling of stalk                               stalk

**Figure 13.** Simulation models of the soil particles, tiger-nut tubers, and stalk [29].

### 4.2. Setting the Simulation Parameters

The simulation parameters include intrinsic parameters and contact parameters of materials. The former could be directly measured through bench tests while the latter was calibrated by combining bench tests and simulation tests, thus improving the reliability of discrete element simulation tests [15]. The contact parameters between tiger-nut tubers and steel plates and those between tiger-nut tubers were obtained by calibration [30,31]. Other parameters were determined by referring to published literature [32–34]. On the basis of research by the research group, the contact parameters and mechanical of the selected materials are shown in Tables 4 and 5.

**Table 4.** Parameters of contact coefficients.

| Materials | Collision Recovery Coefficient | Static Friction Coefficient | Dynamic Friction Coefficient |
|---|---|---|---|
| Tiger nut-Tiger nut | 0.48 | 0.10 | 0.34 |
| Tiger nut-Soil particle | 0.49 | 0.25 | 0.42 |
| Tiger nut-Soil particle | 0.62 | 0.07 | 0.25 |
| Tiger nut-Stalk | 0.35 | 0.02 | 0.32 |
| Soil particle-Soil particle | 0.14 | 0.27 | 0.56 |
| Soil particle-Steel | 0.15 | 0.36 | 0.50 |
| Soil particle-Stalk | 0.11 | 0.09 | 0.21 |
| Stalk-Stalk | 0.26 | 0.01 | 0.32 |
| Stalk-Steel | 0.43 | 0.03 | 0.45 |

**Table 5.** Parameters of mechanical parameters.

| Materials | Poisson Ratio | Density (kg/m$^3$) | Shear Modulus (MPa) |
|---|---|---|---|
| Tiger nut | 0.18 | 1230 | 4 |
| Soil particle | 0.26 | 1179 | 1.1 |
| Steel | 0.27 | 7850 | $8 \times 10^4$ |
| Stalk | 0.42 | 241 | 1.0 |

*4.3. Determination of the Type of Soil-Breaking Blades*

Simulation analysis was conducted according to the influences of different types of soil-breaking blades on the soil-breaking performance, tossing performance, and tuber damage rate. Simulation tests were conducted on straight, L-shaped, and bent blades at an advancing velocity of 0.3 m/s as required for harvesting operation under conditions that the penetrating angle of digger blades, simulation time step, and spindle speed were 15°, 20%, and 400 rpm, respectively. The simulation tests are displayed in Figure 14 and the simulation results are shown in Figure 15.

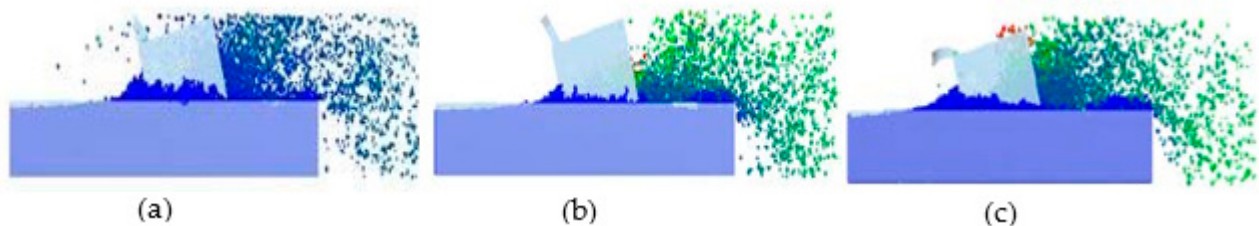

**Figure 14.** Simulation tests on soil-breaking blades. (**a**) Straight blades; (**b**) L-shaped blades; (**c**) Bent blades.

Simulation analysis of different types of blades reveals that straight and bent blades perform well in soil penetration and their soil breaking rates are higher than those of L-shaped blades. Bent blades show the best tossing performance, followed by L-shaped blades, and straight blades exhibit the poorest tossing performance. In the EDEM, the tuber damage rate was measured using the average normal force and the collision times of tiger-nut tubers [35]. The normal force and collision of tiger-nut tubers surge at 2.1 s when using bent blades, which damages the tiger nuts. Therefore, the straight and bent blades were combined when meeting the requirements for soil-breaking performance and tossing performance. The soil-breaking blades on the two sides were two bent blades, while 31 straight blades were used in the middle. These soil-breaking blades showed staggered and balanced arrangement, with a phase difference of 120°, which not only achieve the best soil-breaking rate and tossing performance but also reduced the tuber damage rate.

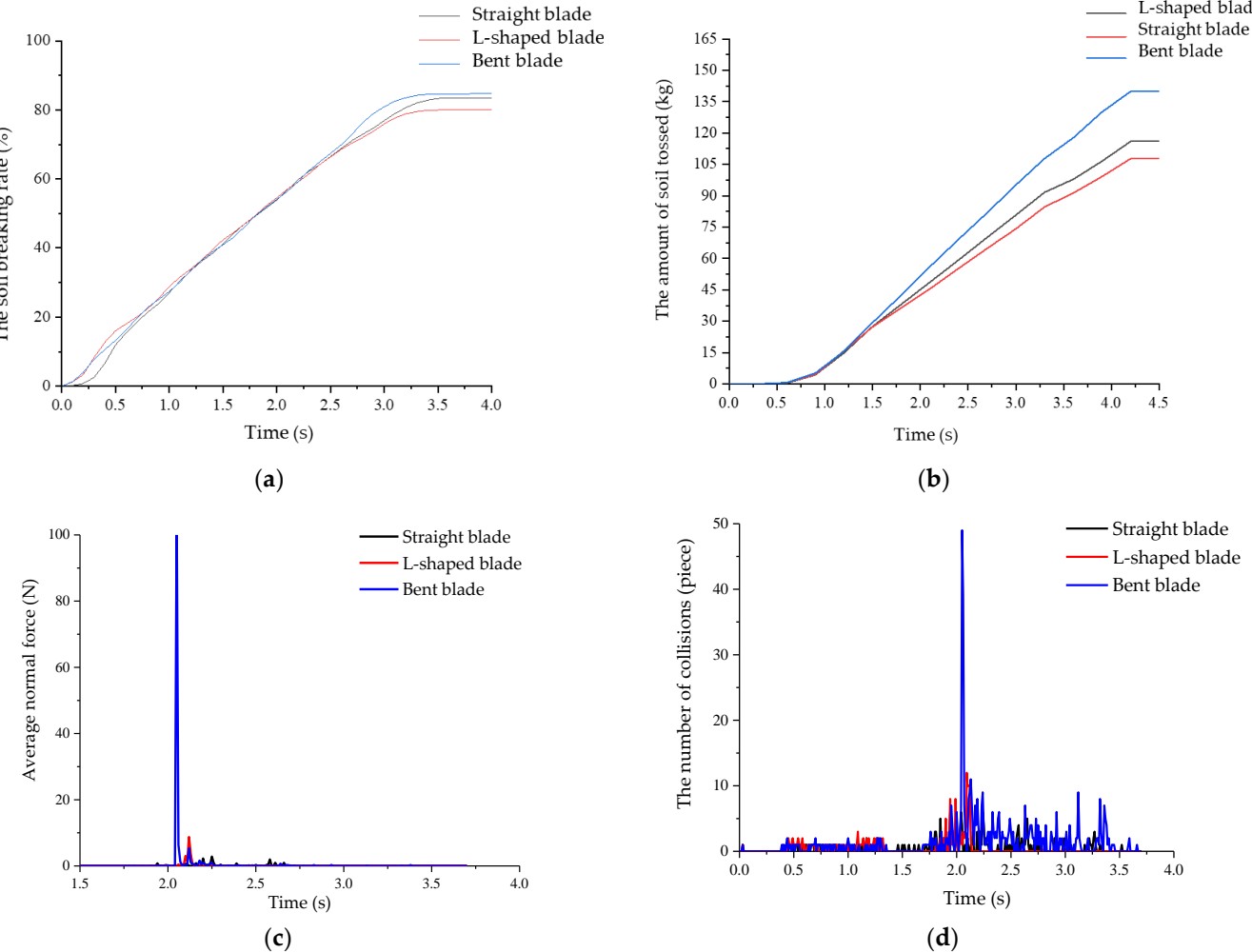

**Figure 15.** Changes in the soil breaking rate, amount of soil tossed, average normal force, and number of collisions with time. (**a**) Changes in the soil breaking rate with time; (**b**) Changes in the amount of soil tossed with time; (**c**) Changes in the average normal force on the tiger nuts with time; (**d**) Changes in the number of collisions between tiger nuts with time.

### 4.4. Simulation and Verification of the Digging and Elevation Process

When harvesting tiger nuts, the mixture of tiger-nut tubers, soil particles, and stalks was tossed to the elevating screen deck by the digging devices through the soil-breaking blades. In contact with the screen deck, the soil blocks were broken due to impact, friction, and rolling. Large soil blocks were broken and sifted through the screen during the delivery of the elevating screen and vibration of the vibrating drum. Based on the EDEM-RecurDyn (joint simulation technology of EDEM and Recursive Dynamic) coupling simulation method, the simulation model for the digging and hoisting devices was established to study the motion of tiger-nut tubers on the elevating screen deck. The motion velocity of tiger-nut tubers reflects the elevation effect of the vibrating hoisting chains. The simulation process and results are shown in Figure 16.

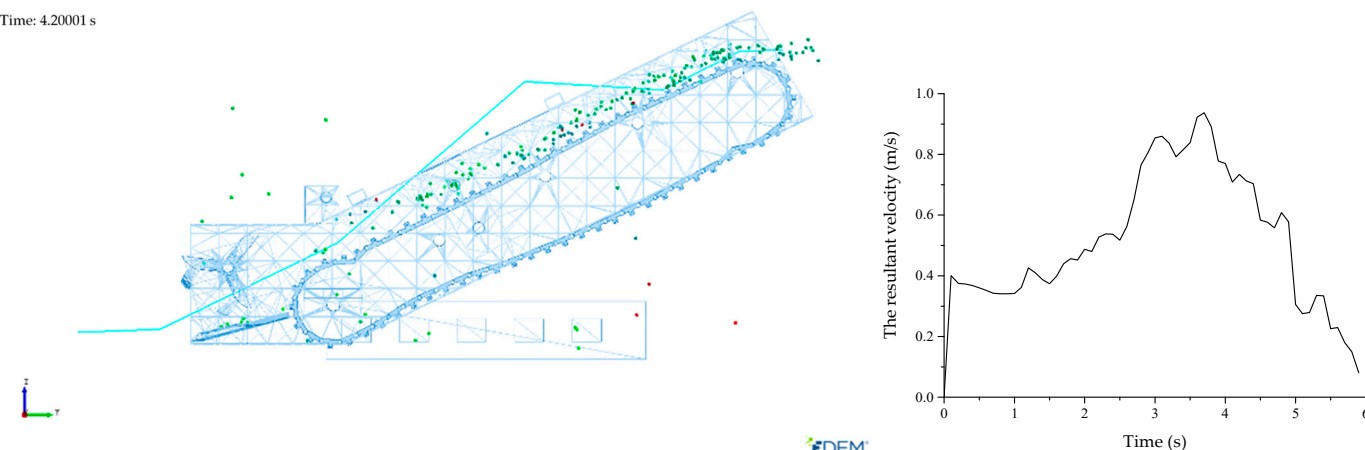

**Figure 16.** Simulation of the motion of tiger-nut tubers and the resultant velocity.

The simulation process implies that tiger nuts in the soil can be smoothly tossed by the digging devices to the hoisting devices and the vibrating drum on the hoisting devices tosses the tiger nuts to a certain height. The changes in the resultant velocity of tiger-nut tubers with time show that tiger-nut tubers are not in contact with the screen deck in the first 2 s and their velocity is approximate to the harvesting speed; after 2 s, tiger-nut tubers begin to make contact with the screen deck and the resultant velocity increases. Only when the velocity exceeds the harvesting speed can tiger nuts tossed by the digging devices be smoothly delivered backward, which reduces plugging in the harvesting process and verifies the efficacy of the designed digging and hoisting devices. That is, the digging and hoisting devices can smoothly carry out both the digging and elevation processes.

### 4.5. Simulation and Verification of the Tuber Picking and Screening Process

When regarding the restitution coefficient and collision strength of extractions were unchanged, different feed rates (FRs) (40, 50, and 60 kg/s) were adopted to perform simulation tests on the tuber picking and screening process under a screen vibration amplitude of 3.98 mm, crank rotational speed of 236.51 rpm, and inclination angle of the screen deck of 6.7° (Figure 17) [13]. The screening efficiency under different FRs is shown in Figure 18.

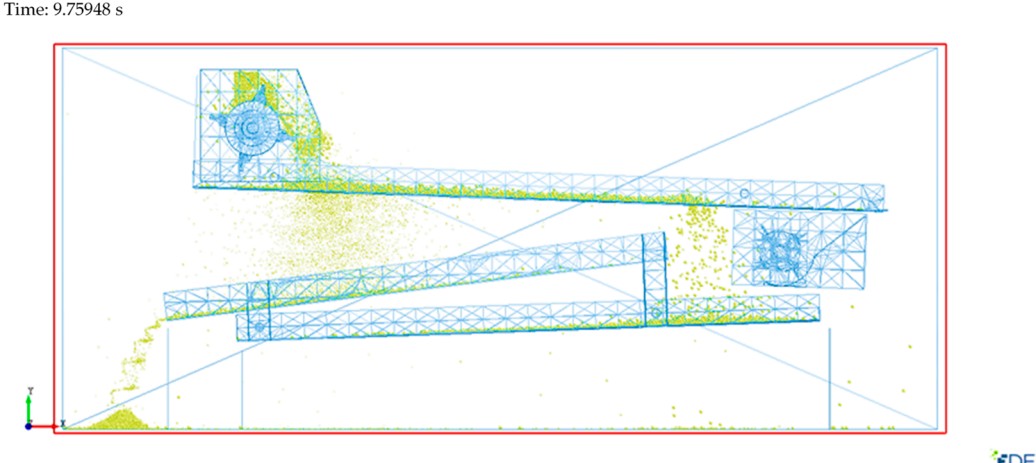

**Figure 17.** Simulation of the tuber picking and screening process.

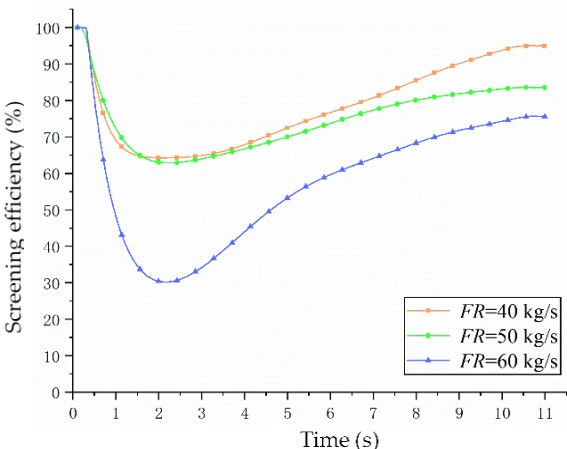

**Figure 18.** Changes in screening efficiency.

It can be seen from Figure 18 that many tiger-nut tubers are accumulated in the front of the screen deck in the initial period of screening, so that small particles cannot come into contact with, or be sifted through, the screen deck. With the vibration of the screen, the particles move to the rear of the screen and small particles fall on and are sifted through the screen deck, which gradually improves the screening efficiency until reaching a stable state. The screening efficiency under three different FRs always decreases at first, then increases, finally reaching a stable value, indicative of the normal operation of the screen. Moreover, the lower the FR is, the higher the screening efficiency and the quicker the stable value is reached. This conforms to the law of motion and verifies the efficacy of the designed tuber picking and screening device.

## 5. Field Tests

After prototype manufacture, the rack, crawler devices, working devices, and other key components were produced or purchased. Then, the whole machinery was assembled, tested, calibrated, and debugged. Finally, field tests were conducted in a tiger nut planting base in Xinzheng City, Henan Province, China in October 2021, as shown in Figure 19.

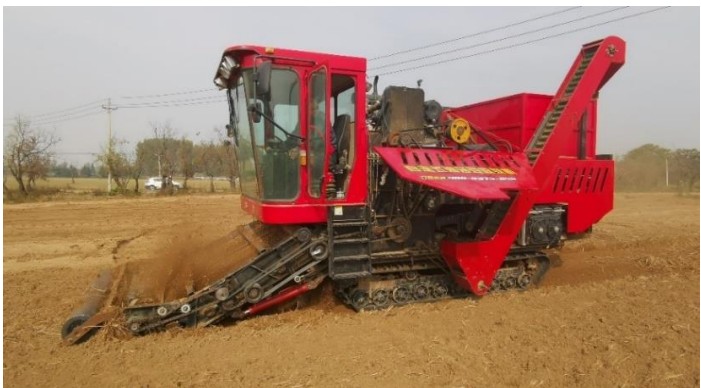

**Figure 19.** Field tests of the harvester.

### 5.1. Test Conditions

Field tests were conducted following the Measuring Methods for Agricultural Machinery Testing Conditions—General Rules (GB/T 5262-2008) [36] and the test devices included an electronic balance, a tape, a steel rule, manual screens, a GPS measurement and positioning device, a TYY-2 soil hardness meter, and a YT-SW soil detector. The five-point sampling method and the random sampling method were utilized to measure the soil properties and tuber growth conditions many times at the test site, and the test results were averaged. The soil moisture content, bulk density of soil, soil compactness, yield of tiger

nuts per mu (an area unit, one mu is equivalent to 667 m$^2$), and tuber growth depth were 12.31%, 1179 kg/m$^3$, 226 kPa, 1370 kg, and 95 mm. The soil was a sandy loam, and the deepest tubers grew at a depth of 120 mm.

### 5.2. Test Methods

The tiger nuts were harvested using the harvester at a digging depth of 130 mm. The operation section was 50 m long, and the 30 m in the middle (with stable operation) was selected as the test section. A total of five harvesting tests were conducted.

The harvesting efficiency was calculated according to the time taken by the harvester to pass through the test section and the area in stable operation using the following equation:

$$E_C = \frac{3600(X \times B)}{10,000T} \tag{23}$$

where $X$ and $B$ represent the length of the test section (m) and the harvesting breadth (m) respectively; $T$ is the time taken by the harvester to pass through the test section (s); $E_C$ denotes the harvesting efficiency (ha/h).

After the harvester passed through the operating section, the tiger nuts, stalks, and soil in the collecting box were separately weighed. After harvesting, three areas of 1 m$^2$ were selected at random from the test section, where the tiger nuts (excluding those with a radius of less than 4 mm) exposed to the ground and buried in the soil were collected and weighed. On this basis, the harvest rate and impurity rate were calculated using the following equations:

$$T_1^* = \frac{W_1}{W_1 + W_2} \times 100\% \tag{24}$$

$$T_2^* = \frac{W_3}{W_1 + W_3} \times 100\% \tag{25}$$

where $T_1^*$ and $T_2^*$ separately denote the harvest rate (%) and the impurity rate (%); $W_1$ represents the mass of tiger nuts in the collecting box (kg); $W_2$ stands for the mass of tiger nuts exposed to the ground and buried in the soil (kg); $W_3$ denotes the mass of impurities in the collecting box (kg).

### 5.3. Test Results

Test results of the crawler-type tiger-nut harvester in the five groups of repeated tests are listed in Table 6. The comparison of the test prototype, reference prototype, and standard requirements is shown in Table 7.

**Table 6.** Field test results.

| Serial Number | Harvesting Efficiency (ha/h) | Harvest Rate (%) | Impurity Rate (%) |
|:---:|:---:|:---:|:---:|
| 1 | 0.208 | 98.37 | 3.38 |
| 2 | 0.216 | 98.30 | 3.11 |
| 3 | 0.201 | 99.21 | 2.79 |
| 4 | 0.230 | 97.45 | 3.57 |
| 5 | 0.225 | 97.37 | 3.35 |
| Average | 0.216 | 98.14 | 3.24 |

**Table 7.** Comparison of test results.

| Test Index | Test Prototype | Reference Prototype [37] | Standard Request [38] |
|:---:|:---:|:---:|:---:|
| Harvesting Efficiency (ha/h) | 0.216 | 0.180 | 0.1 |
| Harvest Rate (%) | 98.14 | 97.10 | ≥95 |
| Impurity Rate (%) | 3.24 | 3.60 | ≤5% |

According to the test results, the average harvesting efficiency, average harvest rate, and average impurity rate of the tiger-nut harvester during stable operation are separately 0.216 ha/h, 98.14%, and 3.24% under conditions wherein the digging depth is 130 mm. The minimum harvesting efficiency is 0.201 ha/h in the five groups of tests, which meets the requirement for harvesting efficiency. The harvest rate and impurity rate tested in the five groups of repeated tests both meet the requirements for harvest rate and impurity rate in the quality evaluation indices. The harvester shows favorable harvesting quality on the whole and can satisfy the harvesting requirement imposed among tiger-nut growers, and compares with the reference prototype, the purpose of improving the harvesting efficiency and quality is achieved.

## 6. Discussion

In previous research on tiger-nut harvesters in China, existing harvesters are traction and drum-type machines, which call for much labor and assistance in the harvesting process, resulting in their low harvesting efficiency [39,40]. In addition, the collected tiger nuts need to be screened again, so they fail to meet the requirement for mechanized harvesting in industries pertaining to tiger nuts. The crawler-type tiger-nut combine harvester developed in the research integrates the digging, elevation, picking, screening, and collection of tiger nuts. Tests reveal that the harvester shows high harvesting efficiency and quality, and the damage of the crawler-type self-propelled harvester to the ground is smaller than the traction-type harvester. In particular, it prevents damage to underground tiger nuts in unharvested farmland.

A literature review shows that the straight, bent, and L-shaped blades are most commonly used in harvesting underground tuber crops. Simulation tests were conducted on the soil-breaking effect, tuber damage, and soil tossing performance of different types of soil-breaking blades in the design. In this way, the optimal combination of blades was determined, which could decrease the tuber smashing rate in the operation of the harvester, and also the favorable soil breaking and tossing performance provides assistance in the subsequent tuber picking and screening stages.

Key devices were designed by combining theoretical analysis and structural design by investigating the planting condition and parameters of tiger nuts. Moreover, verification simulation was conducted on the digging and elevation process and the picking and screening process to ensure that the whole operation process meets the design requirement and conforms to the law of motion. Finally, the prototype was manufactured, and field tests were conducted to verify the harvesting efficiency and quality of the developed tiger-nut combine harvester. The results are all within the range of the design objectives and provide a new research direction for the design of tiger-nut combine harvesters.

When designing key devices of the harvester, we referred to design ideas of self-propelled grain harvesters and crawler-type corn combine harvesters [41,42]. The tuber picking device added between the digging and hoisting devices and the screening device was the horizontal axial flow drum commonly used in grain harvesters, while threshing elements such as the spike teeth, plank teeth, and their arrangement were designed specifically for tiger-nut tubers.

Despite these innovations, the developed crawler-type self-propelled combine harvester for tiger nuts also has the following limitations:

(1) The developed tiger-nut harvester is specific for tiger nuts planted in sandy loam in Xinzheng City; subsequent research should explore the optimization of key devices under different soil structures and consider the applicability and reliability of the machinery in different working environments, so as to meet the requirement of improving the overall applicability and universality;

(2) The developed harvester is based on traditional design to meet the design requirements. An intelligent control system will be studied in future research to develop an adaptive control system with sensing, decision-making, and control functions, to realize the intelligent harvesting of tiger nuts.

## 7. Conclusions

(1) The designed tiger-nut combine harvester is self-propelled; it consists of digging and hoisting devices, a tuber picking and screening device, and crawler devices. It can achieve one-pass integrated harvesting of tiger nuts including the digging, soil removal, picking, screening, and collection thereof, which improves harvesting efficiency and quality and compensates for shortcomings of mechanized harvesting of tiger nuts in China;

(2) The key devices comprising the harvester were theoretically designed, and the combination of straight and bent blades was determined to serve as the soil-breaking blades through simulation analysis of the soil-breaking rate, tossing performance, and tuber damage arising from the use of different types of blades. The stress state of tiger nuts in the elevation process was studied. Results show that when the hoisting angle is $15°$, the normal elevation of tiger nuts is ensured when the elevation speed ranges between 0.63 and 0.81 m/s. The principle of the tuber picking and screening process was analyzed to facilitate the design of the tuber picking drum, double-deck heterodromous vibrating screens, and soil-crushing guide roller. The screening area of the whole harvester reaches 13.6 $m^2$. By calculating and evaluating the supporting capacity of the tracks, the crawler devices were designed on the premise of meeting the requirement for a specific ground contact pressure. The motion and screening efficiency curves of tiger-nut tubers on the hoisting devices were obtained through discrete element simulation. The results indicate that the law of motion conforms to the objective law, thus verifying the rationality of the theoretical design;

(3) Field tests of the prototype show that the harvest rate, impurity rate, and harvesting efficiency of the designed harvester are 98.14%, 3.24%, and 0.216 ha/h when the digging depth is 130 mm. The whole harvester works in a stable manner and various indices all meet the technological requirements imposed by tiger-nut growers in China.

## 8. Patent

Qu, Z.; He, X.; Wang, W.Z.; Zhou, Z.; Lv, Y.L.; Guo, H.Q. Caterpillar Self-propelled Tiger-Nut Harvester and Harvesting Method of Tiger Nut: ZL202011123909.8[P]. 11 December 2020.

**Author Contributions:** Conceptualization, Z.Q. and M.H.; methodology, W.W. and X.H.; investigation, Z.Z. and Y.L.; resources, Z.L.; data curation, M.H.; writing—original draft preparation, Z.Q. and M.H. All authors have read and agreed to the published version of the manuscript.

**Funding:** This research was funded by the special fund for National Key R&D Program of China (Grant No. 2019YFD1002602) and Henan Province Science and Technology Research (Grant No. 212102110217).

**Institutional Review Board Statement:** Not applicable.

**Informed Consent Statement:** Not applicable.

**Data Availability Statement:** The data used to support the findings of this study are available from the corresponding author upon request.

**Acknowledgments:** The authors would like to thank their college and the laboratory.

**Conflicts of Interest:** The authors declare no conflict of interest.

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
