# Peer review of "Design and Test of a Crawler-Type Tiger-Nut Combine Harvester"

_agriculture, doi:10.3390/agriculture13020277_

Round 1

Reviewer 1 Report

Hello!

I carefully analyzed the article received for review, the article is interesting and the graphics help a lot, the images are clear and conclusive.

The introduction is well structured but can be improved with more citations from the field.

Chapter 2 is well structured and clear, but it needs a little arrangement on the page, line 88 - 2.3 Working Principle - can be moved to the next page, same as line 169.

Figure 9 (a) and (b) contain two lines, perhaps they would be better united in a single image.

Table 2 (line 274) is on 2 pages, I recommend finding a solution and moving it to one page, as well as table 3.

The title of figure 15 must be below the figure on the same page.

Subtitle 5.2 can go to the next page.

In Discussions, I did not see any reports or comparisons with results obtained by other authors....bibliographic references must be added.

The WOS or DOI must be added to the References to facilitate access to the articles.

Reviewer 2 Report

Attach comments and suggestions on the manuscript.

Reviewer 3 Report

What are the machine operating parameters? There is no information about this There is only test data.

Round 2

Reviewer 2 Report

Most of the issues mentioned have been improved.

However, I still see problems in some places.

For example, the performance of the reference prototype in Table 7 already satisfies the standard, but it is difficult to understand what the intention of this study is to improve it. Therefore, it seems necessary to properly organize the originality and novelty of this study.

I think these examples are what make this manuscript feel more like a report than a article. Therefore, I think that if the author supplements these parts, it will be a better article.

Reviewer 3 Report

1)All units mentioned in the article should be written with a superscript.

For example kg/m3 must be kg m-3. All units in the article should be corrected in this way

2)how did you measure static and dynamic friction of coefficient?

3) Some values need to be determined experimentally for the simulation. What experimental practices were used to find these values?

4) how did you find Poisson Ratio, Density and Shear Modulus. There is no any information about it

5)there is no result section, i could not see it in article

6)there is no any discusion in discusion section. The results obtained in the article should be compared positively or negatively with the results obtained in different studies and these results should be discussed mutually, but in this article, the discussion section is written in the form of a story. If there is no discussion in a scientific article, this article cannot be considered scientific.

7) The Conclusion section is too short for such a comprehensive article and needs to be expanded.

8) What are the devices used for the tests?

9) Where the material, method and result sections of the article begin and end is not clear in the article.

Round 3

Reviewer 3 Report

When I look at the authors' responses to my second report and the corrections in the manuscript, I see that all corrections were made according to the referees' warnings. With this correction, it is appropriate to accept the publication as it stands.
I wish the authors every success for their future work.